# Caregivers of Patients with Hematological Malignancies within Home Care: A Phenomenological Study

**DOI:** 10.3390/ijerph17114036

**Published:** 2020-06-05

**Authors:** Isabella Capodanno, Mirta Rocchi, Rossella Prandi, Cristina Pedroni, Enrica Tamagnini, Pierluigi Alfieri, Francesco Merli, Luca Ghirotto

**Affiliations:** 1Department of Hematology, Azienda USL–IRCCS di Reggio Emilia, Viale Risorgimento, 80-42123 Reggio Emilia, Italy; Isabella.capodanno@ausl.re.it (I.C.); pierluigi.alfieri@ausl.re.it (P.A.); francesco.merli@ausl.re.it (F.M.); 2Hospice “Casa Madonna dell’Uliveto” Via Oliveto, 34-42020 Albinea, Reggio Emilia, Italy; mirta.rocchi@madonna-uliveto.org; 3Servizio Infermieristico Domiciliare, Azienda USL di Modena, piazzale dei Donatori di Sangue, 3-41012 Carpi, Italy; rossellaprandi@yahoo.it; 4Direzione delle Professioni Sanitarie Azienda USL-IRCCS di Reggio Emilia Viale Amendola, 2-42122 Reggio Emilia, Italy; cristina.pedroni@ausl.re.it; 5Department of Primary Care, Azienda USL-IRCCS di Reggio Emilia Viale Amendola, 2-42122 Reggio Emilia, Italy; enrica.tamagnini@ausl.re.it; 6Qualitative Research Unit, Azienda USL-IRCCS di Reggio Emilia Viale Umberto I, 50-42123 Reggio Emilia, Italy

**Keywords:** caregivers, home care services, hematologic neoplasms, palliative care, qualitative research

## Abstract

The role of caregivers in homecare settings is relevant to the patient’s wellbeing and quality of life. This phenomenon is well described in the literature for the oncological setting but not specifically for that of hematological malignancies. The aim of this study was to explore the experience of primary caregivers of patients with hematological malignancies within home care. We conducted a phenomenological study based on interviews with 17 primary caregivers of hematological patients. Analysis of the contents led to the identification of five main themes. Perhaps, the innovative aspects of this study can be summarized in three points: This service was demonstrated to fulfil the ethical aspects of providing the patient with a dignified accompaniment to the end of life. Secondly, the efficiency of the service and the benefit are directly dependent on the caregivers’ wellbeing, so knowledge of the dynamics and emotions involved can lead to the development and implementation of programs for hematological malignancies. Lastly, a collaborative caregivers–professionals relationship can improve a sense of accomplishment for all parties involved, lessening the family’s frustration related to not having done their best. Home care brings significant benefits for both the patient and the caregivers and fulfils the ethical obligation of providing the patient dignified end-of-life care.

## 1. Introduction

Home care in the oncological setting has demonstrated advantages in terms of quality of life (as perceived both by patients and physicians), lower hospital admissions, and cost of care [1,2,3,4,5,6,7]. Despite its widespread implementation in the oncological setting for patients affected by solid tumors, home care (HC) is scarcely adopted for hematological malignancies (HMs), where it is limited to the end of life when the patient’s condition turns severely critical [8,9,10,11,12,13]. While we can speculate that this may be linked to the unpredictable and uncertain disease course that prevents a definition of the final stages of HMs, it also appears that hematologists seem to maintain their focus on cancer-related treatment goals rather than palliative care and manifest skepticism on the value of palliative care, which they often view as an interference to cancer treatment [14,15,16,17,18]. Nonetheless, home care may play an important role also in the setting of HMs.

In general, patients with HMs are extremely complex and fragile due to both the disease and treatment-related symptoms. Most often, they are older individuals [19,20] with pre-existing comorbid conditions and functional limitation, which further aggravate the disease and treatment burden: Being severely immunocompromised, they are especially susceptible to developing serious infections, hemorrhagic complications (in the occurrence of thrombocytopenia), and variable asthenia depending on the severity of anemia and tolerance to treatments [21,22]. Taken together, this and scarce social and economic resources eventually delay their access to hospitals and increases their need for frequent hospital admissions [23,24,25,26,27].

In fact, elderly patients with HMs are more likely to die in hospital or in hospice, rather than at their home [28,29,30]. In many cases, these patients may also manifest psychological fragility resulting from the lack of relations with family members or other dear persons for extended stays in preventive isolation [31,32,33]. In such a context, for an opportunity for home care, the caregivers (CGs) that interact with both the patients and healthcare professionals (HPs) who rely on their support cover a pivotal role [34].

Gaining a greater understanding of family CGs’ needs and experiences would enhance the overall process of caring for patients with HMs assisted at home. To our knowledge, very little information is available in the literature on CGs in home care settings [34,35,36,37,38,39,40,41,42,43,44,45,46,47,48,49,50], especially in advanced HMs. Although in several studies, CGs describe positive outcomes, such as feelings of privilege, accomplishment, and improved family relationships [34], other studies pointed out the heavy burden perceived by the CG and the physical and emotional exhaustion experienced especially at the end of life [35,37]. In fact, family CGs report the weight of high levels of responsibility, isolation, and anxiety, as well as emotions, such as anger, fear, frustration, and sadness [35,37,48,50]. The experience of CGs of patients with terminal illness assisted at home is often characterized by fatigue [37,38,39]. In this regard, the literature reveals a global fatigue that comes from witnessing the patients’ suffering, pain, and death. For some CGs, the responsibility of caring for and making choices instead of the patient causes fatigue as well.

Some of the most important issues are a lack of exhaustive communication between HPs and families [37], advice by HPs, support and training during home assistance [36], and an empathic attitude towards CGs [41]. Some studies report that CGs’ expectations for support sometimes had apparently not been met by HPs and they felt themselves viewed as ‘co-workers’ instead of people with their own needs [36,37,38,39,40].

CGs need support and relief from the health professionals, both in organizational (i.e., management of therapies) and psychological terms. This broad and complex support aims at creating the indispensable conditions to make CGs’ feel supported. Other studies highlight the importance of giving the CGs specific information and training about how to identify and manage possible incidents or complications in relation to the disease, the therapy, and the supportive/palliative care [41].

The aim of this work was to explore the question: ‘What is the experience of primary CGs of patients with hematological malignancies within home care?’

## 2. Materials and Methods

We adopted a descriptive phenomenological study [51], which allowed an investigation of the subjective meaning of the participants’ lived experiences and provided insight into people’s motivations and actions [52,53]. An important aspect of the descriptive approach in phenomenology requires researchers putting aside past knowledge, bracketing all possible pre-assumptions [54] to encounter and describe the essential characteristics of the experience studied.

### 2.1. Setting and Sampling

The study was conducted within the Reggio Emilia Hematological Home Care (HHC) program [55], which cooperates closely with family CGs to make home care feasible. The HHC team is made up of a hematologist and a nurse, both with experience and training in the field of palliative care and are responsible for ensuring the specialized care of patients with HMs, especially those who are significantly immunosuppressed or unable to access hospital care.

Sampling was performed purposively among all primary CGs who had a loved one admitted to the HHC who passed away due to an HM at least 6 months earlier, was Italian or English speaking, and was willing to talk about their experience. Exclusion criteria were cognitive impairment and an incapacity to express consent. The hematologist and the nurse identified and selected information-rich cases and then made the initial contact with each CG. The participants were subsequently approached by telephone by the interviewer to set a time and place for the meetings. The interviewers (L.G., M.R.) were a qualitative methodologist, with a background in social science research, and a nurse, trained in palliative care working in a hospice, with no former experience in HHC. The interviewers had no previous contacts with the participants. CGs also received information sheets outlining the study and complete privacy and confidentiality information. If the participant requested it, the interview could be conducted in the presence of a loved one. A summary of the home care program offered in the Reggio Emilia health district is briefly described in Table 1.

### 2.2. Data Collection

Data were collected between June 2016 and July 2018, through open-ended interviews. The interview guide (Table 2) focused on the experience of assisting the loved one at home and related emotions or insight. As suggested by Patton [56], researchers defined specific foci (experience, feelings, and knowledge) and related open-ended questions to allow participants to provide rich descriptions of their own experience. We chose to ask only questions that supported participants’ story telling. We did not follow any theoretical framework in defining the interview guide.

### 2.3. Data Analysis

All the interviews were transcribed verbatim. No participant desired to read or comment on the transcript of their own interview. Interview contents were analyzed using Colaizzi’s methodological indications [51], divided into six analytical steps. (1) L.G., M.R., R.P., and C.P. read the interviews several times to get a sense of participants overall experiences; (2) M.R. and R.P. identified the significant statements and shared them with L.G. and C.P., who gave a third opinion; (3) four analysts formulated and validated through discussion the meanings of each significant statement, helping each other to exclude pre-conceived or biased meanings’ description. During this phase, in particular, L.G:, M.R., R.P., and C.P. reflected about possibilities; (4) M.R. and R.P. organized each significant statement into meaning units and sub-theme into major themes; (5) L.G. checked the meaning units, sub-themes, and themes; and (6) all researchers agreed on the final definition of the overarching statements to summarize the participant’s lived experience. Analyses were done collaboratively and managed using purposely structured Microsoft Word tables. No computer data analysis software was used.

### 2.4. Bracketing, Reflexivity, and Rigor

To describe participants’ experiences from a research perspective as impartially as possible, we managed to bracket preconceptions of the phenomenon by: (i) Posing open-ended questions during the interview to increase the likelihood that participants would elaborate and share experiences; and (ii) involving a research team of four members, coming from different disciplines and background, in the data analysis process (intersubjective corroboration) [54]. Personal reflections on themes were shared, discussed, and compared to the findings of this study.

To ensure rigor, an external audit was conducted by I.C., F.M., P.A. (hematologists), and E.T. (nurse), who checked the interview transcripts and analysis. Credibility and originality of the analysis were obtained by gathering rich in-depth data from interviews, and by transcribing verbatim and analyzing line by line using the participants’ own words as much as possible [57]. Finally, the study report followed the standards for reporting qualitative research (SRQR) [58].

### 2.5. Ethical Considerations

Ethical approval was obtained from the Reggio Emilia Provincial Ethics Committee (Prot. n. 2016/12381 of 2016/05/17). All the participants provided informed written consent for the interviews and to enable publication of the extracts.

## 3. Results

The study considered 19 CGs; 2 of them refused to participate as they were still felt too emotional to tell their story. In total, 14 CGs accepted the invitation (three of them wanted to be accompanied by a close relative). The final sample consisted of 17 CGs whose characteristics are detailed in Table 3. Data were collected through 14 in-depth interviews (mean duration about 46’, ranging from 23’to 70’).

The analysis yielded five core themes: (a) Feeling connected with the patient; (b) experiencing the path with fatigue; (c) perceiving home care as an opportunity; (d) feeling the support from HPs; and (e) developing a sense of self-efficacy. Themes and sub-themes are outlined in Table 4.

### 3.1. Feeling Connected with the Patient

This theme describes the CGs‘ feelings resonating with the patient’s status: Feeling relief and joy when their loved one appeared energetic, well, determined, or strong; and feeling concern and sorrow when the patient was sad or aware of the severity of his/her situation.

“C1: Once she called me and she asked me: ‘give me morphine, there is pain, I feel pain’[…] Since she clearly said that she wanted to die… for us it was just […] she continuously ... ‘I want to die, I want to die, I want to die’ […].

C2: Because she said ‘a child should never see the mother in this…’ so it had become ... a very heavy situation” [13PPMM19].

CGs were aware that their presence was appreciated by their loved one, which made them feel gratified and relieved.

“He is very close to me, he has always said: ‘I come to your house’ [...] he wanted to be with me and be followed by me, he felt calm” [3PF2].

The connection with the patient was also reflected by the strong commitment of the CG to participate in defining the care path. CGs respected the patient’s will of being informed and always avoided discussing death with him/her.

“So, I never did anything she wouldn’t want to do … That way instead, she didn’t have any…, I didn’t have anything to regret myself. I’m happy it went that way …” [9PF111].

According to CGs, all the family members who lived at home participated in the loved one’s journey at home.

“For them (grandchildren) it surely was a moment of growth, seeing their grandfather facing the disease this way” [1PF6].

### 3.2. Experiencing the Path with Fatigue

This category describes the perceptions of fatigue experienced by family members. The effort to protect the loved one was evident. CGs defined their relatives as very fragile and dependent on the care and protection they provided. Most often, this led to a sense of fatigue. Some CGs also felt the burden of being invested of the responsibility of caring for their loved one. In some cases, the CGs expressed regrets that emerged posteriori regarding their accompaniment to the patient.

“Because I had to walk him to the bathroom, at night-time, too, because I was afraid, she would fall. So, she would get up 2, 3 times each night. I was tough...” [14PPFM4].

“I always regretted that I couldn’t have been with her more. My mum said: ‘stop it because you’ve been there anyway’” [2PF9].

CGs had centered their lives completely around the patient, concentrating all their energies on the relative, adjusting their own habits and silencing their emotions and their own needs in the background.

“I was able not to cry in front of him except for once, but he would say “You’re making me feel worse”, so I had to keep everything inside, you know?” [6PF15].

Witnessing the suffering of the loved one (as well as attending at his/her death) was painful and in some cases led the CGs to perceive the need to distance themselves from the pain of their loved one. Acknowledging such a need further increased the burden with a sense of guilt.

“I could not go near the bed ... I sat there near the door on the chair because I could not go near him. I didn’t feel up to it” [8PF15].

Besides, the presence or help by other family members throughout the caring path was perceived by the CG as providing some relief.

“My sister-in-law stayed the afternoons, because, you know, I could no longer stand staying there the whole day” [4PPFF15].

### 3.3. Perceiving Home Care as an Opportunity 

This theme collects the meanings CGs gave to the HHC program. Participants perceived HHC as an opportunity to maintain the caring process as intimate and respectful. At the same time, this provided a sense of satisfaction, making them feel they were doing things right.

“My mother had been lucky, she never stayed a day in the hospital except for her children’s birth, so this allowed to preserve an intimate dimension around her illness” [5PM3].

Participants stated that HHC respected the private everyday dimension of the patient’s lifestyle as well as that of the entire family. All saw the value of the “usual normality” on a patient’s well-being, while also providing the opportunity to cultivate family relationships under challenging times. 

“I continued to live my usual life, with just two people at home, one of which severely ill” [3PF4].

The home was seen as a privileged and sometimes indispensable place to enable the involvement of family members in the patient’s end-of-life experience. The desire and the pleasure of sharing all the events with the patient was expressed by several CGs.

“To end in one’s own home represented quality to us...You know, at least you have the dignity and peace to suffer unbothered without or having interferences or concerns when you’re there suffering with people around” [5PM9].

CGs felt that their dedication and the presence of a nurse and a hematologist, at home, were like a gift. Sometimes, caring activities were seen as giving back, a demonstration of gratitude to their loved one.

“I would do it again over and over, I don’t know how many times, since he gave me so much and I really owed it to him. And then anyhow, every now and then would even joke about it with him” [4PPFF10].

### 3.4. Feeling the Support from HPs

This category describes the perceptions related to HHC. Many participants felt HHC as a natural continuation of the path started in the hospital when the worsening of the disease made it difficult even to go to the doctor alone. From a more practical point of view, CGs perceived the convenience of a service that came into action when the patient’s ambulation worsened, making typical daily actions difficult. Not having to go to the hospital for visits meant saving the tribulations associated with onerous travel in terms of time and effort. With home care, CGs could continue to make one’s life at home while awaiting medical help.

“I would say that homecare started almost immediately because she wasn’t capable of…we’d take her around in her wheelchair, but it was tiring for her, so…there was the option to have homecare, so we accepted and welcomed it gladly” [12PM1].

Participants reported that HPs provided them relief, in terms of organization (technical, management of therapies), training (perceiving an education by HPs), and psychological support. HPs tried to help CGs in taking on their new role. This support was also repeatedly mentioned, highlighting the exceptional professionalism of the HPs. Relational modalities and empathy were considered intrinsic elements of such professionalism, which established a relationship of trust and reliance with HPs.

All participants reported the prompt availability of HPs at all times, which provided them reassurance even when the care path became steep and uncertain.

“P: Sure, they taught me what to do and whenever I encountered some difficulties, I had their number to call them to receive immediate reassurance, some support, so I never had problems” [3PF6].

CGs spoke about the family environment favored by the relationship with HPs. CGs appreciated the average length of time spent by HPs and their positive and smiling attitude, which gave a sense of familiarity.

“Well, he (the patient) was so happy, also because they were so kind, and they had come to establish a very good and friendly relationship” [8PF3].

### 3.5. Developing a Sense of Self-Efficacy

This theme gathers the experiences of undertaking the CG role with determination. Often, participants reported their firm belief that they were responsible for the loved one. This attitude took the form of timely and effective choices and actions in order to take control of the situation.

“We got organized because the wheelchair, the thingy…the armchair we brought was one of those that you can put upright, a mobile armchair, not like those rigid ones… and then the air conditioning...it was so hot, that’s another thing we set up in that period to make her comfortable at home, that also made a difference” [7PF18]. 

The sub-theme of spirit of initiative depicts the CG’s confidence and determination towards taking charge over their loved one’s care path, with a spirit of initiative and dedication, which make home management feasible. Many interviewees revealed the satisfaction of “feeling capable”, especially in the assistance acts, but also in finding solutions and knowing how to organize and manage situations. 

“We did simple things, we thought there would be huge changes, but we found practical solutions in no time” [5PM12].

Analyses of behaviors and through words showed that CGs rarely questioned their responsibility and capability of being a carer.

“We never gave up until the end, though…I think we did the right thing” [1PF6].

## 4. Discussion

The experience of CGs providing assistance to a relative with a HM is intense and complex. CGs devote a great deal of energy and time in caring for their loved ones in the most respectful and dignifying way despite moments of high and lows. They take great responsibility upon themselves, while fearing not being able to cope with the psychological burden of fatigue and suffering. While in other studies family caregivers reported the weight of high levels of responsibility, isolation, and anxiety, as well as emotions, such as anger, fear, frustration, and sadness [35,37,41], one of the core themes in our study is the CGs’ sense of self-efficacy. Our findings regarding the experience of the path with fatigue are consistent with those of other studies that have examined the experience of caregivers of patients with terminal illness assisted at home [37,38,39]. Besides, our participants experienced fatigue and emotional distress because they felt connected to the loved ones: They witnessed their suffering and pain and had to cope with their loss.

Nonetheless, HHC preserved the normalcy and guaranteed the private, domestic, and everyday dimension of living. While the role of the home environment in nurturing the sense of normalcy in patients’ and caregivers’ lives has been discussed in other studies [38,43,44], the perception of home assistance as an opportunity to maintain an intimate and respectful privacy is a novel aspect.

The presence and availability of home-care HPs was another important aspect, as it reassured CGs, easing their sense of inadequacy in providing care, and loneliness in bearing responsibility. Most CGs were pleasantly surprised by the comprehensiveness of the service, as compared to their previously low expectations. Finally, the literacy activity provided by HHC alongside home care empowered the caregiver with skills and notions to overcome practical difficulties and be more aware and capable of responding to the patients’ needs.

Other studies report that CGs’ needs for support sometimes have not been met by HPs [36,37]. CGs may feel they are treated like ‘co-workers’ instead of persons with their own needs [38,39,40]. In our study, CGs felt support and relief from the HPs. Our participants reported they received emotional support and proper information for dealing with the management of therapies. Accordingly, the importance of providing CGs with specific information and training about how to identify and manage possible incidents or complications has emerged from the literature [41]. Furthermore, many studies stress that the HPs’ empathic attitude towards informal CGs is essential to validate CGs’ emotions and encourage them to express them [39,41,42]. What Quiñoa-Salanova and colleagues [41] note in relation to the hospital setting play a pivotal role also within HHC: relational modalities, sensitivity, and humanity by HPs. Another important aspect perceived by the CGs is the prompt availability of HPs: This experience is contrary to what is reported elsewhere [36,37,38,39,40,41,42,43,44,45,46,47,48,49].

This resulted in a better sense of accomplishment and morale boost among CGs. Altogether these elements contributed to the patient’s natural transition to palliative care and to a dignified end of life in a peaceful environment, as opposed to the public hospital environment.

Overall, our findings appear consistent with other cancer settings. As to novel aspects, it describes the great potential home care could cover in the HM setting. One interesting aspect is the comments made by participants on the empathy showed by HPs. This was very much appreciated by the CG and made them feel respected as a person with feelings and opinions of their own, rather than someone who was being delegated to execute a set of tasks on behalf of HPs.

It is beyond doubt that the wellbeing of the CG is considered somewhat marginal respect to the patient’s physical and psychological experience, and the detrimental effects of assisting a loved one alone are largely underestimated. Nonetheless, given the epidemiology of HM and the number of family CGs, CG wellbeing should not be dismissed but rather be considered as an integral part of the patient’s path of care and addressed appropriately.

Accordingly, future HHC programs should be designed to include (i) literacy initiatives for CGs to provide them a basic understanding of disease progression and symptom management; (ii) training of HPs on the role of soft skills in home care, such as practicing empathy and providing psychological relief and reassurance towards the CG; and (iii) establish a direct phone line for CGs to communicate in real time with HPs providing home care.

### Strengths and Limitations

The present study is the first to our knowledge describing the experience and needs of a heterogeneous sample of family CGs (husbands, wives, sons, and daughters) assisting patients with HMs at home during end of life, since palliative care is sparingly suggested to HM patients, at least in our country. The findings provide insight on a peculiar disease setting, where there appears to be strong potential for improvement. It is noteworthy to mention, however, that our findings stem from a single center experience and are not associated to a standard protocol of care and therefore might not be generalizable to other healthcare districts or countries. We can hypothesize that those who accepted to be interviewed were more likely to be satisfied with the HHC program (selection bias) and may have given positive-oriented responses (also for the social desirability bias). Researchers did not perform member checking of the results. As a qualitative study, the results are not generalizable to other populations, although they offer a useful reference point for other research among CGs of patients with HMs in other cultures and contexts.

## 5. Conclusions

This study provides a deeper understanding of the lived experience of individuals who are the CG of a relative assisted at home for an advanced HM. This is an important finding, since only few studies to date have explored the role and experiences of family CGs in patients with HMs [35,41,46,47,48,49,50], and none of them referred to a setting of home assistance for patients with advanced diseases.

The study also clarifies the importance of considering CGs as a person having care needs in their own right, and not as substitute healthcare providers: The CGs’ needs should sometimes be considered equally with respect to the patient’s needs. When this occurs, for many CGs, it is a source of great help and relief.

A high level of professionalism, humanity, and a prompt availability of the HPs in case of need are considered essential elements for the proper functioning of hematological home care. The fact that our participants were satisfied with the home care and support they had received from healthcare staff suggests that the path is feasible and the HHC team was working well.

## Figures and Tables

**Table 1 ijerph-17-04036-t001:** Reggio Emilia health district hematological home care service program.

**The hospital hematologists wrote guidelines for the management of blood and platelet transfusions, administration of intravenous and subcutaneous chemotherapy, and the management of clinical complications before the HHC service started. The HHC team involves one hematologist trained in palliative care and one nurse with expertise in the management of hematological patients and palliative care. Both cooperate with hospital professionals. The HHC service operates jointly with primary care doctors.**
**Screening**
The hospital hematologists recommend the HHC service to patients based on the following criteria: Patient living in the urban area covered by the HHC service; Physical limitations and/or compromised clinical and personal status (with at least two of the following conditions: ECOG performance status ≥ or Karnofsky Performance Status ≤ 50%; unable to walk and/or without a person to accompany him/her to the hospital; high risk of infection); Adequate venous access; as to the home environment: suitable and safe for HHC service, adult caregiver, capable, collaborating and helpful caregiver at home (family member, friend or home-aid assistant). Other criteria regard the caregiving situation: caregivers’ cohabitation with the patient (the presence of a helpful person during the time needed for assistance is required); availability/possibility to be absent from work for assistance; health status of the caregiver; architectural barriers and impossibility to modify the home of the patient according to new needs; uncleanliness and unhealthy or poorly ventilated location; the presence of pets.
**Activities (abstract from guidelines)**
The staff should provide all the required information, both during the first visit at home and during the assistance. In particular, the nurse provides training moments about hygiene practices, assessment of vital parameters, caring of venous catheters and urinary catheters, wound dressing, management and prevention of constipation. For severe cases (terminal illness, symptoms as bleeding, dyspnea, vomit, breakthrough pain, and delirium), caregivers are trained to administer drugs, based on anticipatory prescriptions. The hematologist can be contacted by telephone 24 h a day. Chemotherapy offered comprises subcutaneous and intramuscular injections and intravenous treatments with one or two drugs. The hematologist and the nurse remain at the patient’s home throughout the chemotherapy infusion

**Table 2 ijerph-17-04036-t002:** Interview guide.

Foci	Exemplifying Question(s)
*Starting*	“Thank you for participating. I would like to ask you if the reason we are here is clear? Are there any questions you wish to ask me? Are there any doubts that you want me to clarify?”
*The experience of returning to domestic life after the hospital discharge*	“Could you please tell me what you thought when healthcare professionals offered home care to your loved-one?”
*The emotional experience of assisting the loved one at home*	“Could you please tell me how you experienced the care of your loved one? What was your experience of assisting your loved one?”
*The experience of intra-family organizational changes*	“Could you please tell me how was your daily life? Could you please tell me how your organization was?”
*Actions related to the end of life of the patient*	“Could you please tell me what happened at home in the last few days?”
*The emotional aspects related to the end of life of the patient at home*	“Could you please tell me how you did feel about managing your loved one’s last moments at home?”
*Conclusion*	“Is there any other though you would like to share? Anything to add?”

**Table 3 ijerph-17-04036-t003:** Participants’ characteristics.

Code	Age Range	Occupation	Education	The CG is the Patient’s	The CG was
1PF	61–70	Retired	Secondary	Wife	Alone
2PF	61–70	Teacher	Tertiary	Daughter	Alone
3PF	41–50	Employee	Secondary	Daughter	Alone
4PPFF	61–70	Unemployed	Primary	Wife	Helped by the daughter
4PPFF	31–40	Employee	Secondary	Daughter	Helped by the mother
5PM	41–50	Sport trainer	Tertiary	Son	Helped by the brother
6PF	61–70	Retired	Tertiary	Wife	Helped by children
7PF	51–60	Un-employed	Secondary	Daughter	Helped by siblings
8PF	50	Un-employed	Secondary	Wife	Alone
9PF	51–60	Employee	Secondary	Daughter	Helped by in-home assistant
10PM	61–70	Retired	Secondary	Husband	Helped by the daughter
11PF	41–50	Employee	Secondary	Daughter	Helped by the mother
12PM	41–50	Employee	Secondary	Son	Helped by the father
13PPMM	31–40	Researcher	Tertiary	Son	Helped by the father
13PPMM	61–70	Retired	Secondary	Husband	Helped by the son
14PPFM	<71	Retired	Primary	Wife	Helped by the son
14PPFM	51–60	Employee	Secondary	Son	Helped by the mother

**Table 4 ijerph-17-04036-t004:** Results’summary.

Main Themes	Sub-Themes
*Feeling connected with the patient*	Feeling the strength of the loved-one
	Accepting the patient’s wishes
	Participating the care
*Experiencing the path with fatigue*	Feeling the burden
	Neglecting themselves
	Needing relief
*Perceiving home care as an opportunity*	Preserving normal life
	Strengthening family bonds
	Making a gift to the loved-one
*Feeling the support from HPs*	Perceiving home care as necessary
	Feeling the professionalism and prompt availability
	Comfort and sense of familiarity
*Developing a sense of self-efficacy*	Spirit of initiative
	Building determination

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
