# Peer review of "Caregivers of Patients with Hematological Malignancies within Home Care: A Phenomenological Study"

_ijerph, 2020, doi:10.3390/ijerph17114036_

Round 1

Reviewer 1 Report

This research contributes to increasing knowledge about the study phenomenon. However, I believe that, in order to be published, authors must improve several fundamental aspects. The first, related to the qualitative methodology used to improve the transparency of the research process and thus ensure the transferability of the study results. The second, related to the results in which the narratives have narratives that do not clearly reflect the authors' description of the different topics. The third related to ethical aspects of the research. The fourth related to the study discussion and study limitations.

Next, I will break down each of these aspects:

In line 51, the authors indicate that it is a descriptive phenomenological study. However, they do not indicate what is their previous position and / or their previous beliefs about the study phenomenon in order, on the one hand, to comply with the provisions of the standards for reporting qualitative research SRQR (S6 - assumptions, and / or presuppositions), and on the other, to establish the necessary bracketing in qualitative descriptive phenomenological studies. Therefore, I think that a necessary question that the study authors must rethink is whether it is a descriptive phenomenological study or an interpretive phenomenological study since they do not indicate that bracketing has been carried out and the study categories of the interviews (Table 2), according to the information provided in the study, suggests that they have been previously established by researchers based on their beliefs about the study phenomenon or the previous theoretical framework from which they start. In addition, whether it is one type of study or another, they must describe what is the criterion used for the construction of the interviews and the pre-established research areas and why they decide to ask these questions and not others.

In line 104, the study authors indicate “No participant desired to read or comment the transcript of their own interview”. However, they do not indicate whether the member checking of the interpretations of the results by the researchers has been carried out. They must indicate whether the member cheking has been carried out, how it has been carried out. If it has been carried out, they must indicate whether there were discrepancies between the interpretation of the results by the researchers and the experiences reported by the participants during the member checking.

In line 113, according to the information provided by the authors, it is understood that the analysis of the data has been carried out manually and without the use of qualitative results analysis software. However, they must indicate this explicitly.

In line 119, the authors must indicate what was the role of the researcher's field notes in the triangulation process of data sources and how the researcher's field notes were integrated in the process of analysis of the results.

At line 142, the participant's narrative does not clearly reflect the description of the topic that the authors describe between lines 139-141. To justify this result, a representative narrative that justifies the description must be included.

In line 145, the authors indicate “CGs were aware that their presence was appreciated by their loved-one which made them feel gratified and relieved”. However, they do not provide a narrative that reflects this result.

In line 151, the authors indicate “According to CGs we interviewed all the family members who lived at home participate to the loved-one’s journey at home”. If more participants have been interviewed than previously indicated in Table 3, the characteristics of those participants should be described. On the other hand, before interviewing these participants… did they sign the informed consent? The claim made by the authors suggests that members of the CG family who have not signed the informed consent may have been interviewed and violate ethical criteria of the investigation. If so, all results obtained from interviews with these participants should be removed from the study.

In line 159, the authors indicate “In some cases, the CGs expressed regrets that emerged posteriori regarding their accompaniment to the patient”. However, they do not provide a narrative that reflects this result.

In line 168, the authors indicate “Witnessing the suffering of the loved-one (as well as attending at his/her death) was painful and in some cases led the CGs to perceive the need to distance themselves from the pain of their loved-one. Acknowledging such need further increased the burden with a sense of guilt”. However, they do not provide a narrative that reflects this result.

The narrative in line 177 does not reflect any previously described results. The way of presenting the results must be consistent with the rest of the document, first describing the topic or subtopic and then accompanying it with the corresponding narrative.

Section “4. Discussion” (line 247) is incorrect. The authors include a summary of their results and indications for the development of proper health care for these people, but they do not discuss their results with studies of international relevance. I encourage the authors of the study to review the correct way to prepare a discussion based on studies published in international journals and to rewrite this section.

Section “4.1. Strengths and limitations ”(line 278) does not include the limitations of the qualitative methodology in which the results are not generalizable to other contexts, and the social desirability, in which the participants could have given positive responses about the care received with the in order to please the interviewers.

Reviewer 2 Report

The article produce important information related to the experience of primary caregivers of patients with hematological malignancies within home care.

I have some suggestions to the manuscript:

The title does not correspond to the research question. It needs clarification.

The introduction is clear. However, I suggest that the authors open more such issues as: what kind of challenges caregivers face in daily care in practice? How those challenges influence caregiver’s everyday life?

Tables 3 and 4 need clarification.

Discussion is quite narrow. There is a need for more versatile reflection based on results and earlier literature.

Round 2

Reviewer 1 Report

The authors have responded satisfactorily to the comments made during the previous review, so I consider that the manuscript can be considered for publication.